# Closed for business: The mortality impact of business closures during the Covid-19 pandemic

Dion Bongaerts[1], Francesco Mazzola[1]*, Wolf Wagner[1,2]

**1** Rotterdam School of Management, Finance Department, Erasmus University, Rotterdam, The Netherlands,
**2** Centre for Economic Policy Research, London, England

* mazzola@rsm.nl

## Abstract

We investigate the effectiveness of business shutdowns to contain the Covid-19 disease. In March 2020, Italy shut down operations in selected sectors of its economy. Using a difference-in-differences approach, we find that municipalities with higher exposure to closed sectors experienced subsequently lower mortality rates. The implied life savings exceed 9,400 people over a period of less than a month. We also find that business closures exhibited rapidly diminishing returns and had large effects outside the closed businesses themselves, including spillovers to other municipalities. Overall, the results suggest business shutdowns are effective, but should be selectively implemented and centrally coordinated.

## Introduction

In attempts to contain pandemics, policy makers trade off public health benefits against economic costs. Yet, little is known about how containment policies should be designed or how they should be organized. Countries have varied widely in their responses to the outbreak of the novel coronavirus (SARS-CoV-2). Some governments have closed almost their entire economy for certain periods of time, while many others have chosen selective closures of varying breadth. In some countries, policies were predominantly decided upon at the regional level, whereas in others policies were determined at the national level. However, containment interventions were not necessarily instigated by public authorities. For example, drug gangs in favelas in Rio de Janeiro imposed the first lockdowns in Brazil. As measures are selectively reversed and reintroduced, there has been a recent tendency for more and more localized approaches, often going hand-in-hand with decentralization of decision-making. At the extreme end, laissez-faire-like approaches (such as the one followed by Sweden) fully decentralize decisions by relying on individual businesses and households to curb the virus's spread.

The empirical examination of containment policies is challenging for several reasons [1]. Containment measures are typically implemented in response to a rapidly evolving pandemic. Therefore, there are no clear counterfactuals to these policies—the spread of the virus would also have changed in the absence of such policies. Containment decisions are also often clustered, making it difficult to isolate the impact of a specific policy. Policies tend to be introduced

**Data Availability Statement:** The data sets used in this study are available at Figshare (figshare.com), as follows: - Daily deaths across Italian municipalities in 2020 in excess with respect to 2015-2019: 10.25397/eur.14500491 - Italian

Municipality characteristics affecting Covid-19 contagion: 10.25397/eur.14500449.

**Funding:** The author(s) received no specific funding for this work.

**Competing interests:** The authors have declared that no competing interests exist.

around the time the general public becomes acutely aware of the dangers of a virus, and takes measures, often self-imposed, to reduce the risk of contracting the virus.

By exploiting within-country variation in exposure to a nationwide containment policy, this paper examines the health impact of business shutdowns during the 2020 Covid-19 pandemic. Italy was the first European country to shut down parts of its economy. Specifically, on March 11$^{th}$ the Italian Government first shut down sectors comprising 17.6% of the economy (in terms of employment). On March 25$^{th}$ additional sectors comprising 32.5% of the economy were closed. Importantly, this "one-size-fits-all" closure policy affected Italian municipalities differently due to heterogeneity in sectoral exposures. Our empirical strategy is to study differential changes in mortality patterns across municipalities, allowing to control for confounding factors taking place in Italy around the time of the policy.

The results suggest that business shutdowns are effective in saving human lives: municipalities with higher exposures to the first shutdown see a decline in mortality rates relative to other municipalities. Based on our estimates, we undertake a counterfactual analysis that shows the first shutdown saved about 9,500 Italian lives over 24 days. Estimates of the Value of a Statistical Life (VSL) years imply a large societal benefit from business closures that exceed nine billion Euros. Interestingly, we find the second shutdown is not as effective (per unit of employment closed down) as the first one, supporting the notion that the first shutdown had already reduced the spread of the virus.

Our analysis also shows that business shutdowns have important mortality spillovers. Mandated business closures might affect the spread of the virus outside a municipality because of commuting (other forms of travel were fairly restricted during our sample period). Consistent with this hypothesis, we find that shutdown exposure in business centres affects mortality rates in neighbouring municipalities. The beneficial effect is large, and comparable in size to the impact of a municipality's own shutdown exposure. We also find that greater shutdown exposure has a strong effect on parts of the population that is very unlikely to be working, e.g., the elderly. This points to significant contagion effects that reach beyond the firms undertaking economic activities. The existence of different forms of spillover suggests a need to coordinate shutdown decisions at a central level.

Finally, our analysis points at rapidly declining benefits to scale from sectoral shutdowns. We compare the marginal effectiveness of shutdowns across municipalities that differ with respect to the proportion of their economy that was affected by the first shutdown. We find the marginal effectiveness in municipalities with the lowest sector exposure to be about three times higher than the average marginal effectiveness across all municipalities. Since the marginal effectiveness also varies across sectors (we find per-unit shutdowns in some sectors to be more effective than in others), our evidence points broadly in favour of targeted closures of a select number of sectors, rather than more uniform shutdowns.

In the wake of the Covid-19 crisis, a significant theoretical literature is emerging that examines optimal policies during a pandemic. This literature emphasizes *production externalities* as a rationale for public policies. Production externalities arise when the provision of goods and services results in the spread of the virus to individuals not directly involved in the business activities. As firms (and their workers) will not internalize the social cost of contagion, they will make inefficient containment decisions, providing a need for government-imposed shutdowns. To the best of our knowledge, our paper is the first to document empirical evidence consistent with such production externalities. [2] study optimal taxation of business activities (which can be interpreted as shutdown intensity) in an environment where the severity of the production externalities varies with the spread of the virus. Our finding—of lower policy effectiveness once the pandemic is more under control—is consistent with the theoretical premises of their model. [3] show that individuals shifting activities to environments that pose less

contagion risk mitigate production externalities. Our estimates—which are net of such miti-
gating behaviour—suggest that production externalities remain significant. Calibrating an
SIR-model to the US economy [4], show that the social cost of infections exceeds the private
cost by factor two, which is consistent with our results of large spillovers on individuals that
are unlikely to work. [5] analyse virus contagion across jurisdictions (countries, in their
model) and show that inter-jurisdictional externalities create a need to coordinate contain-
ment policies. Our findings of strong geographical spillovers provide empirical evidence for
the existence of such externalities.

The various government approaches taken in response to pandemics are spurring a rapidly
evolving literature that tries to understand their benefits, as well as their costs. Relative to the
benefits side, several papers have examined the impact on mobility, infections and/or ulti-
mately on mortality. Using predominantly time-series and/or cross-country variation, these
studies have generally concluded that government interventions are effective. See, for example,
[6] for France; [7] for Germany; [8] for China; [9–12] for the US; [13, 14] for a counterfactual
analysis for Sweden; [15, 16] for a cross-country study. For a collection of working papers, see
[17]. Our study, using within-country variation in exposure to national business shutdowns in
Italy, confirms and quantifies the effectiveness of containment measures. In addition, we show
that business shutdowns have first-order spillover effects as well as declining marginal returns.
These results provide valuable information for policymakers in their ongoing challenge of
devising and adapting containment policies.

## Materials and methods

### Main analysis

On March 11[th] 2020, the Italian Prime Minister mandated to shut down all food, retail and
personal-services activities. Businesses such as supermarkets, small grocery shops, pharmacies,
and newsstand kiosks were allowed to remain open. The national decree also introduced
restrictions on personal mobility. Two weeks later, on March 25[th], the list of sectors included
in the shutdown policy was enlarged.

For reasons of identification, our study primarily focuses on the first policy. We study the
impact of this shutdown on mortality rates using a difference-in-differences (hereafter diff-in-
diff) approach with continuous treatment. Given the lag between a virus infection and a (possi-
ble) subsequent death, the "treatment" date does not coincide with the day the policy was
enacted. It is impossible for the policy to have any effect on mortality from its implementation
date. Estimates in the literature suggest a median of five days between exposure to the virus
and the occurrence of first symptoms [18] and about eight days between first symptoms and
death [19]. Thus, the median time to death across individuals is about 13 days. Due to variation
around this mean, any effective policy will arguably produce its effects two or three days before
this date; we therefore take the "treatment date" to be day 10.

Our empirical model takes the following form:

$$
\begin{aligned}
y_{m,t} = {} & \beta y_{m,t-1} + \gamma d11_t + \phi(d11_t \times Shutdown11_m) + \\
& + \eta d25_t + \psi(d25_m \times Shutdown25_m) + \theta(d11_t \times Controls_m) + FE_{m,tsa,t} + \varepsilon_{m,t}
\end{aligned}
\tag{1}
$$

where $y_{m,t}$ is the (Covid-related) mortality rate in municipality $m$ on day $t$. We include the
lagged value, $y_{m,t-1}$, as a determinant since epidemiological models (such as the SIR model,
[20, 21]) show that new infections condition highly on the prevalent share of infected people
in the population. We include two dummies, $d11_t$ and $d25_t$, to indicate the treatment date
for the first and second shutdown, both taking value of one as of the tenth day after their

respective announcements. The variables $Shutdown11_m$ and $Shutdown25_m$ measure the exposure of a municipality to sectors that were incrementally shut down at the first and second shutdown, respectively. Our variable of interest is the interaction coefficient $\phi$, which captures whether municipalities with a higher shutdown exposure experience lower daily mortality rates as a consequence of the policy. If the policy shutdown is effective at reducing the spread of the virus, and thus ultimately reduces mortality, then the prediction is that the coefficient $\phi$ will enter with a negative sign. We saturate the model with municipality and day-fixed effects as well as with proxies for a municipality's stage in the pandemic ($tsa$; to be explained below). Lastly, we include interactions of municipality-level variables with the policy dummy $d11_t$ to control for heterogeneous patterns across municipalities following the policy that are unrelated to business shutdown exposures (e.g., due to confounding policies).

We next describe the calculation of the variables. We construct measures of shutdown exposure using granular data on employment and establishments of Italian firms made available by the Italian Statistical Agency (ISTAT). The dataset provides sectoral data at the municipality-level from the year 2017 [22], including information on the number of employees and business owners, revenues and number of establishments. We construct a continuous municipality-level shutdown exposure $Shutdown11_m$ by dividing the number of employees and business owners in sectors shut down on March $11^{th}$ by the total number of all employees and business owners in the municipality. Shutdown sectors correspond to the following European classification of the economic activities (NACE) codes: "451", "452", "473", "474", "477", "478" for the retail industry; "561", "563" for the food and beverages industry; "96" for the personal-services industry. We exclude employment in schooling and sports (NACE codes "85" and "931") from the denominator, since these sectors were already shut down weeks before. Additionally, we exclude from the sample the eleven municipalities in northern Italy that were quarantined as "red zone", since the shutdown policies we focus on do not apply there. We construct an equivalent measure of the second shutdown, $Shutdown25_m$, using the employment ratio of sectors that were incrementally shut down on the $25^{th}$.

We next describe our measure of policy effectiveness, which is based on Covid-19 deaths per 100,000 inhabitants. In early June 2020, ISTAT released death registry data for 7,272 Italian municipalities (covering more than 93% of the entire population, see [23]). The dataset contains the number of deaths per day, along with the residence location, gender, and age bracket of the deceased, over the first quarter of 2020. Using mortality rates offers several advantages. First, alternative measures of Covid-19 related outcomes based on infections or hospital admissions suffer from biases. For example, a higher Covid-19 test intensity will inevitably show higher infection rates. In addition, in regions with worse healthcare conditions and low proximity of hospitals, the usage of hospitals per capita was (mechanically) lower. Many deceased people who had shown no or mild symptoms (asymptomatics) were simply not accounted in the official Covid-19 statistics because they were not hospitalized (e.g., due to the limited capacity of hospitals). Second, the collection process for death registry records minimizes reporting lags and subjectivity in recording information (e.g., residence at time of death). [24] show there was also significant underreporting of official Covid-19 deaths in Italy. Third, deaths also capture mortality cases that are indirectly attributable to Covid-19. For example, evidence suggests that mortality resulting from heart attacks more than tripled during the pandemic in Italy [25], likely because of hospital congestion or the unavailability of ambulances.

A disadvantage of the death registry data, though, is the information missing on the cause of death. We therefore use a statistical method to infer deaths related to Covid-19, based on deviations from historical patterns. Specifically, we calculate excess mortality, i.e., attributable to Covid-19, by deducting from a municipality's (daily) number of deaths the average number

of deaths over the previous five years in the same municipality, using an evenly-spaced-around window of seven days. We scale excess deaths by the population to arrive at the following measure:

$$ExcDeathRate_{m,t} = \frac{Deaths_{m,t,2020} - avgDeaths_{m,7d,2019-2015}}{Population_m} \times 100,000 \qquad (2)$$

There is an important source of heterogeneity across municipalities: the virus reached different municipalities at different points in time. Failing to address this heterogeneity is likely to lead to an inappropriate econometric specification. In particular, a municipality that was hit early by the virus might likely display lower growth in contagion (as the curve has already levelled off) compared to a region with low virus intensity. Due to the highly non-linear dynamics of a pandemic, municipality-level fixed effects might not appropriately account for such heterogeneity. In our empirical analysis, we therefore account in our empirical analysis for the "time of arrival" of the virus to compare municipalities that are at the same stage of the pandemic. We classify the time of arrival in a municipality based on two criteria: "anomaly" and "persistence". The former is measured by the day in which the cumulative excess deaths in a municipality surpass one standard deviation of its distribution over the first four months of 2020. For the latter criterion we require that the cumulative mortality rate among residents of a municipality $m$ reaches a threshold of 100 deaths per 100,000 inhabitants at some point during the same sample period. We classify the time of arrival of a virus as the day when the first criterion is met for a municipality that fulfils the second criterion (which is time-invariant). We have visually inspected our classifications for a number of municipalities, and have found them to be reasonable. Studying the econometric properties, we find the arrival day of the virus lies in between the first and second structural break of a municipality's cumulative mortality rate time series.

Note that according to our definition, some municipalities were not subjected to the virus during the first four months of 2020 (about 500 municipalities). We exclude these from our main dataset, but use them later in a placebo test. To limit noise in our measure of Covid-19 deaths per capita, we also require a municipality to have at least 4,500 inhabitants. This leaves us with 2,145 municipalities, spanning 105 provinces and 20 regions. The sample covers the period from February 22$^{nd}$ to April 13$^{th}$.

Table 1 provides the summary statistics of our sample. The mean across time and municipalities of the variable $ExcDeathRate_{m,t}$ is about 4. That is, there are on average four Covid-related deaths a day per 100,000 inhabitants. The average exposure to the first shutdown, $Shutdown11_m$, is about 17.6%, whereas the average exposure to the second policy, $Shutdown25_m$ is larger (32.4%). There is also sizable cross-sectional variation in the exposure to both policies. As explained, we will focus mostly on the first shutdown; the impact of the second shutdown might be partly confounded by the first one and hence offers a less clean setting. The table also contains information on the breakdown of sectors closed on the March 11$^{th}$. We can see that both the food and beverage sector and the retail sector on average are about 7%, whereas the personal services sector is smaller (less than 3%).

The table also lists several other variables that are used in the analysis. To control for heterogeneity in characteristics that affect transmission, virulence, or the ability to comply with movement restrictions, we collect data on population density ($PopDensity_m$), education ($HighSchool_p$), income inequality ($IncIneq_m$), age structure ($Elderly_p$), and the degree of internal commuting ($IntMob_m$). The $Hospit_p$ variable is the hospital capacity in a province $p$, measured as the sum of the number of beds available in hospitals, as a fraction of the total population (source: Ministry of Health). To study spillovers, we include information on the shutdown

**Table 1. Summary statistics.**

| Variable | Mean | Standard Dev. | P5 | Median | P95 | Observat. |
|---|---|---|---|---|---|---|
| $ExcDeathRate_{m,t}$ | 3.570 | 7.429 | 0 | 0 | 17.379 | 101,794 |
| $\Delta ExcDeathRate_{m,t,t-1}$ | -.0686 | 9.156 | -15.113 | 0 | 14.892 | 101,794 |
| $gExcDeathRate_{m,t,t-1}$ | 1.841 | 5.431 | -0.933 | 0 | 14.892 | 101,794 |
| $\log(cExcDeathRate_{m,t}+1)$ | 5.109 | 0.514 | 4.292 | 5.094 | 5.972 | 101,794 |
| $\log(Pop_m)$ | 9.237 | .628 | 8.467 | 9.107 | 10.430 | 101,794 |
| $d11_t$ | .507 | .499 | 0 | 0 | 1 | 101,794 |
| $d25_t$ | .190 | .392 | 0 | 0 | 1 | 101,794 |
| $Shutdown11_m$ | 17.563 | 7.080 | 7.814 | 16.624 | 29.690 | 101,794 |
| $Shutdown25_m$ | 32.378 | 14.345 | 12.894 | 30.314 | 59.168 | 101,794 |
| $Food11_m$ | 7.174 | 4.265 | 2.563 | 6.141 | 15.370 | 101,794 |
| $Retail11_m$ | 7.490 | 3.489 | 2.891 | 7.058 | 13.748 | 101,794 |
| $Personal11_m$ | 2.898 | 1.605 | 1.128 | 2.659 | 5.266 | 101,794 |
| $IntMob_m$ | 40.621 | 12.250 | 24.19 | 38.52 | 62.53 | 101,794 |
| $PopDens_m$ | 5.845 | 1.077 | 4.101 | 5.804 | 7.685 | 101,794 |
| $HighSchool_p$ | 56.37 | 8.22 | 41.93 | 56.52 | 69.66 | 101,713 |
| $IncIneq_m$ | 10.942 | 6.851 | 4.46 | 9.31 | 23.11 | 101,794 |
| $Elderly_p$ | 36.24 | 4.28 | 29.6 | 35.7 | 43.9 | 98,738 |
| $Hospitaliz_p$ | 0.183 | 0.0556 | 0.104 | 0.177 | 0.281 | 101,794 |
| $RelShutdown11_n$ | 23.731 | 5.645 | 16.538 | 23.925 | 32.235 | 97,958 |
| $AbsShutdown11_n$ | 17.265 | 2.824 | 12.808 | 17.023 | 21.549 | 99,958 |
| $PopShutdown11_n$ | 17.064 | 2.803 | 12.808 | 16.962 | 21.533 | 99,942 |
| $WinterTourists_p$ | 5.838 | 2.708 | .273 | 2.145 | 22.418 | 98,477 |
| $WeekArrival_m$ | 5.713 | 2.78 | 1 | 6 | 10 | 101,689 |
| $DaysArrival_{m,t}$ | 37.465 | 18.518 | 8 | 37 | 69 | 101,794 |

This table shows the mean, standard deviation, the 5th, 50th (median) and 95th percentile, and number of observations for each variable used in the empirical analysis. Variable definitions are in S1 Table. Source: [23].

exposures of the largest business centre of the province where the municipality is located. Among the province's larger municipalities (with at least 16,500 inhabitants), we identify the business centre in three alternative ways: as the municipality with the highest relative share of closed sectors, $RelShutdown11_n$; the average shutdown exposure of these municipalities under this method is 23.73%, which is by construction larger than the unconditional mean; next, as the one with the highest absolute number of employees and business owners in the shut down sectors, $AbsShutdown11_n$; finally as the one with the largest population, $PopShutdown11_n$. Next, the variable $WinterTourists_p$ measures tourist intensity in a province. It is calculated as (foreign) tourists visits during January and February, scaled by population [26]. The top-10 provinces according to our tourist proxy contains skiing provinces (e.g., Trento, Bolzano, Sondrio) and historical cities (e.g., Florence, Venice, Rome). The $WeekArrival_m$ variable is the number of weeks that elapsed between the arrival of the virus (calculated as described above) and the effective date of the first policy (March 21$^{st}$). We can see that, on average, a municipality starts experiencing the virus for the first time in early February, about one month before the effective date of the first policy. Lastly, $DaysArrival_{m,t}$ measures the time elapsed from the day on which the pandemic begins in a municipality $m$. A comprehensive definition of the variables can be found in S1 Table.

## Results and discussion

We start with a graphical analysis of mortality rates across municipalities over the sample period. Fig 1 shows excess mortality, comparing municipalities with above and below median shutdown exposure.

As explained, an important source of heterogeneity in the dynamics among municipalities is the stage at which they were first affected by the pandemic. To take this into account, for the construction of the graph in Fig 1 we group municipalities according to the week when the virus was first recorded according to our methodology. Within each of these cohorts, we calculate the average excess mortality rates for municipalities above and below the median shutdown exposure, and then average across cohorts. This ensures that the low and high exposure groups have an equal composition in terms of the stage of the pandemic the municipalities are in. Note that in our regressions, we also do control for the stage of the pandemic. Because such effects are likely to be non-linear in measures of the stage of the pandemic, we do this using granular pandemic stage fixed effects.

There are three takeaways from Fig 1. First, there is no visible difference among the "more treated" and "less treated" groups before the policy is enacted (solid line) and before the policy can become effective (dashed line), both in terms of trends but also in terms of their level. This strengthens the premises of our diff-in-diff analysis. Second, the two groups diverge around the effective date of the first policy. Third, following this effective date, excess mortality rates decline more in high exposure municipalities relative to low exposure municipalities.

Table 2 compares both groups more formally during the pre-treatment period. Panel A investigates the "parallel-trends"assumption, showing that mortality rates in the high and low exposure group do not statistically differ—both in changes but also in levels—prior to the effective date. Panel B of Table 2 does, however, show that treament and control groups *ex ante* differ in terms of population density, income equality, and internal commuting characteristics. Differences in these variables might potentially result in different responses to a common policy, thereby interfering with the diff-in-diff analysis [1]. In our empirical work, we therefore include interactions terms between the post-treatment dummy $d_{11}$ and these variables to control for such effects.

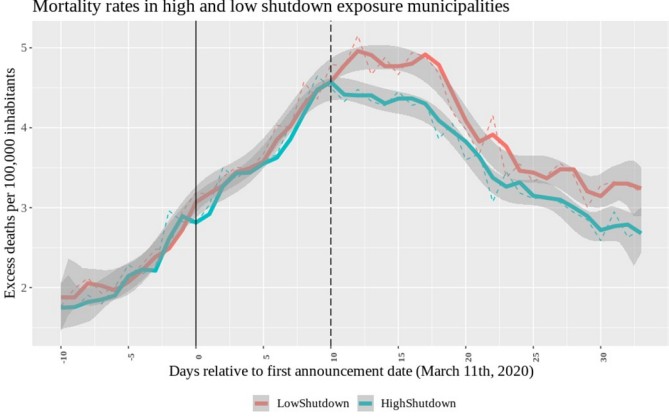

**Fig 1. Mortality rates in high and low shutdown exposure municipalities.** Average within-group excess mortality rates over time. Vertical lines identify the first announcement ($\tau = 0$ corresponds to 03/11; the solid line) and treatment date ($\tau = 10$, corresponding to 03/21; the dashed line). A group excess mortality rate is calculated by averaging excess mortality rates across municipalities with above the median $Shutdown11_m$ (treated, N = 1,076) and those below the median (control, N = 1,069), conditional on pre-sorting municipalities on the virus arrival week. Blue line: HighShutdown. Red line: LowShutdown.

**Table 2. Parallel trend analysis and balanced covariates test.**

| Panel A: Parallel Trend | | | | |
|---|---|---|---|---|
| | Mean | | | |
| | Low Shut11$_m$ | High Shut11$_m$ | Difference | T-test |
| $ExcDeathRate_m$ | 3.521 | 3.499 | 0.022 | 0.122 |
| $\Delta ExcDeathRate_m$ | -0.373 | -0.333 | -0.04 | -0.267 |
| $gExcDeathRate_m$ | 1.758 | 1.683 | 0.0756 | 1.095 |
| **Panel B: Municipality characteristics** | | | | |
| | Mean | | | |
| | Low Shut11$_m$ | High Shut11$_m$ | Difference | T-test |
| $WeekArrival_m$ | 5.464 | 5.595 | -0.131 | -1.067 |
| $IntMob_m$ | 36.345 | 44.530 | -7.985 | -15.962*** |
| $PopDens_m$ | 6.055 | 5.691 | 0.364 | 7.952*** |
| $HighSchool_p$ | 56.474 | 56.219 | 0.255 | 0.714 |
| $IncIneq_m$ | 10.418 | 11.202 | -0.784 | -2.712*** |
| $Elderly_p$ | 36.314 | 36.09 | 0.224 | 1.203 |

Comparison of municipalities with high and low exposure to the first shutdown. Municipalities are first assigned into groups (above and below the median) conditional on their virus arrival week. Values are averaged over the 10-days period surrounding the first policy announcement (03/07 to 03/15). Variable definitions are in S1 Table.

Table 3 contains the OLS estimates for our main empirical model (Eq (1)). The first column reports the results including municipality and days-since-arrival fixed effects. The variable of interest, the interaction term of the treatment dummy and the shutdown variable, obtains a coefficient of -0.0279, which is significant. This indicates the shutdown was effective, as municipalities with a higher share of sectors that were shut down saw their mortality rates decline more relative to other municipalities. The coefficient on the post-treatment time dummy $d11_t$ obtains a positive sign. This is explained by the fact that the policy was initiated in response to information about a rapidly spreading virus, thus around the time where contagion rates where peaking. As previously discussed, this points to an endogeneity in adopting national containment policies, and reinforces the need to use within-country variation for identification. The lagged values of excess mortality rates positively predict next day excess mortality rates, consistent with epidemiological models.

Column (2) also includes the exposure to the second shutdown. The coefficient on the interaction term for the first shutdown increases in (absolute) size, to -0.0453. The dummy for the second shutdown obtains a negative and significant value, consistent with the second shutdown happening at a time of a nationwide decline in mortality rates. The interaction effect with the second shutdown exposure, $Shutdown25_m$, obtains a negative value of -0.0228, which is significant. This suggests the second shutdown was also effective in reducing mortality rates. It is interesting to compare the coefficients for the interaction effects on the first and second shutdown. The first shutdown obtains a coefficient that is about twice as large as the second one. This might indicate declining returns to shutting down sectors, which is an issue we will return to below.

From column (3) onward, we include interactions of the post-treatment dummy $d11_t$ and the variables that showed significant differences between the control and treatment samples in Table 2. In this way, we control for potentially heterogeneous patterns across municipalities unrelated to shutdown exposure. Only the interaction with population density shows up as significant, indicating that mortality rates increase more rapidly in more densely populated areas

**Table 3. Main analysis.**

| LHS: $DailyDeathRate_{m,t}$ | (1) | (2) | (3) | (4) | (5) |
|---|---|---|---|---|---|
| $d11_t \times Shutdown11_t$ | -0.0279*** | -0.0453*** | -0.0437*** | -0.0474*** | -0.0358*** |
| | (-2.75) | (-4.14) | (-4.36) | (-4.72) | (-4.56) |
| $d11_t$ | 1.129* | 0.986** | -0.192 | | |
| | (1.93) | (2.19) | (-0.33) | | |
| $d25_t \times Shutdown25_m$ | | -0.0228*** | -0.0231*** | -0.0223*** | -0.00527 |
| | | (-3.45) | (-3.45) | (-3.37) | (-1.08) |
| $d25_t$ | | -0.0924*** | -0.0929*** | | |
| | | (-3.01) | (-3.02) | | |
| $PopDens_m \times d11_t$ | | | 0.169** | 0.204*** | 0.197*** |
| | | | (2.34) | (2.79) | (3.15) |
| $IncIneq_m \times d11_t$ | | | -0.0100 | -0.0144 | -0.00837 |
| | | | (-1.10) | (-1.58) | (-1.36) |
| $IntMob_m \times d11_t$ | | | 0.00643 | 0.00777 | 0.00978* |
| | | | (1.30) | (1.55) | (1.75) |
| $y_{m,t-1}$ | 0.0754*** | 0.0703*** | 0.0701*** | 0.0659*** | 0.0367*** |
| | (6.53) | (6.42) | (6.41) | (6.14) | (4.07) |
| Municipality FE | ✓ | ✓ | ✓ | ✓ | ✓ |
| ArrivalDay FE | ✓ | ✓ | ✓ | ✓ | x |
| Time FE | x | x | x | ✓ | x |
| ArrDay × Time FE | x | x | x | x | ✓ |
| Obs. | 100,656 | 100,656 | 100,656 | 100,656 | 100,563 |
| $R^2$ | 0.213 | 0.216 | 0.216 | 0.220 | 0.269 |
| Adj. $R^2$ | 0.195 | 0.198 | 0.198 | 0.202 | 0.228 |

This table presents difference-in-differences estimates of municipality-level panel regressions of daily excess mortality rates (left-hand side). The lagged dependent variable $y_{m,t-1}$ is included in the model. $d11_t$ and $d25_t$ are dummy variables that take a respective value of one in the days after the first and second policy becoming effective. $Shutdown11_m$ and $Shutdown25_m$ are employment exposures of municipality $m$ to the shutdown policies of March 11th and March 25th, respectively. The sample consists in ISTAT death registry data over the period 02/22/2020–04/13/2020. $t$ statistics in parentheses. Standard Errors clustered at municipality- and day-level. *, ** and *** represent statistical significance at the 10%, 5% and 1% level respectively. Variable definitions are in S1 Table.

after the policy introduction, which we consider intuitive. Notably the coefficient on the interaction term of the post-treatment dummy and the first shutdown exposure, $d11_t \times Shutdown11_m$, is hardly affected by the inclusion of these controls.

In column (4), we also include day fixed effects and in column (5) we saturate the model with days-since-virus-arrival times day fixed effects, with the latter controlling for any (potentially non-linear) dynamics in mortality arising from municipalities being in different stages of the pandemic. The coefficient on the interaction term of the treatment dummy and the first shutdown exposure continues to be statistically significant. However, the inclusion of days-since-virus-arrival times day fixed effects renders the effect of the second shutdown insignificant.

Our regression results suggest the business shutdowns implemented in Italy reduced mortality arising from Covid-19. The size of the coefficients also suggests the effect is substantial in economic terms. We can obtain an estimate of the total effect of the first shutdown as follows. Given an average shutdown exposure across municipalities of 17.6% and a coefficient estimate of -0.0358 (last column of Table 3), the first shutdown reduced mortality by 15.64 per 100,000 inhabitants over our 24-day sample period. This number is slightly higher than the direct effect obtained by $17.6 \times 0.0358 \times 24$, due to an indirect effect through the lagged dependent variable.

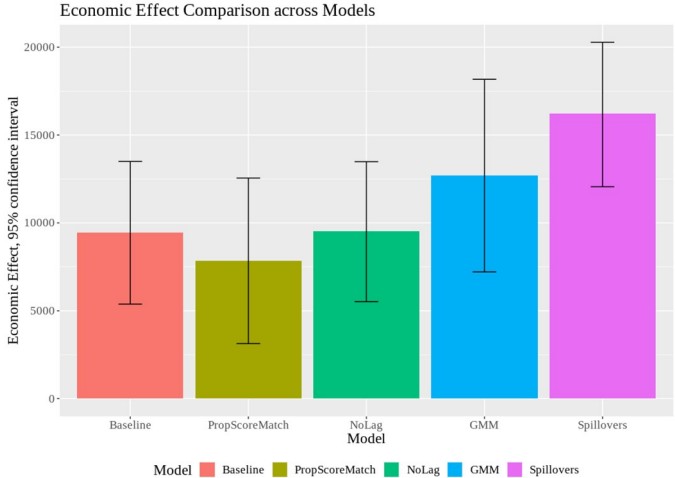

**Fig 2. Economic effect comparison across models.** Economic effects across models. For the back-of-the-envelope calculation, the number of human lives saved (y-axis) is estimated using a 24-days treatment period, the mean shutdown exposure (see Table 1) and a 60.36 million population (except for "Spillovers", which is net of the population in business centres). Segments centered at the top of each bar denote 95 percent confidence intervals of the regression coefficient of interest (i.e., on d11 × Shutd11) of the respective model specification (x-axis). Models: Baseline; PropScoreMatch; NoLag; GMM; Spillovers.

Given a population in Italy of 60.36 million, this totals to 9, 439 lives saved, with a 95% confidence interval between 5,379 and 13,499 (see Fig 2).

Using estimates for the "Value of Statistical Life", we can translate this into monetary terms. A common estimate for the value of one year of life in Europe is €80,000 [27]. Considering 12 years of average remaining life of Covid-19 victims [28], we can calculate the monetary benefit of the policy to be 9,439 × 80,000 × 12 = €9 billion.

## Robustness

Table 4 contains the estimation results from various modifications of our baseline specification (the last column of Table 3).

Our analysis has shown that high and low exposure municipalities do not display differences in the outcome variable (both in terms of level and trend) prior to the effective date. Thus, the standard parallel-trend assumption on which identification in diff-in-diff models relies is met. However, a potential concern is that municipalities with different shutdown exposures might also differ among other dimensions, and that this might create heterogeneous responses to confounding effects around the effective date. To some extent, we alleviate such concerns by including interaction terms of the post treatment date dummy $d_{11}$ and important municipality-level characteristics (in columns 3, 4, and 5 of Table 3). In addition, we perform an analysis in which we match "treated" (above median exposure) to "control" municipalities (below median exposure). Using a two-step propensity score-matching algorithm, we first match on the week of virus arrival and subsequently on the three characteristics that showed up significantly in Table 2 (population density, income inequality, and mobility). We match each treated municipality to a control with replacement, and discard treated municipalities for which no good match is available. This procedure yields in total 873 treated and 444 control municipalities. As expected, there are no longer significant differences between the treatment and control sample when we perform the balance variables test of Table 2 on the matched

**Table 4. Robustness tests.**

| | PropScore Matching | Exclude Lombardy | Exclude Touristic | Shorter Window | Arrival Time | No Lag | System GMM | Placebo | | |
|---|---|---|---|---|---|---|---|---|---|---|
| | (1) | (2) | (3) | (4) | (5) | (6) | (7) | (8) | (9) | (10) |
| $d11_t \times Shutd11_m$ | -0.0300*** | -0.0242*** | -0.320*** | -0.0337*** | -0.316*** | -0.0375*** | -0.0410*** | -0.0154 | -0.00020 | -0.00713 |
| | (-3.26) | (-2.87) | (-3.69) | (-4.28) | (-3.31) | (-4.65) | (-4.54) | (-1.33) | (-0.02) | (0.49) |
| $d25_t \times Shutd25_m$ | -0.0066 | 0.0075* | -0.00564 | | -0.00613 | -0.0055 | -0.0169*** | | 0.00051 | 0.0098 |
| | (-0.97) | (1.72) | (-0.98) | | (-1.15) | (-1.09) | (-2.88) | | (0.10) | (1.50) |
| $y_{m,t-1}$ | 0.0283*** | 0.00225 | 0.0387*** | 0.0225** | 0.0026 | | 0.186*** | -0.0378*** | 0.364*** | 0.0112 |
| | (3.24) | (0.45) | (3.92) | (2.09) | (0.27) | | (2.53) | (-3.19) | (3.97) | (0.75) |
| Interact. Controls | ✓ | ✓ | ✓ | ✓ | ✓ | ✓ | ✓ | ✓ | ✓ | ✓ |
| Municipality FE | ✓ | ✓ | ✓ | ✓ | ✓ | ✓ | x | ✓ | ✓ | ✓ |
| ArrD × Time FE | ✓ | ✓ | ✓ | ✓ | ✓ | ✓ | x | ✓ | ✓ | ✓ |
| Observations | 61,885 | 79,281 | 79,722 | 85,520 | 70,608 | 102,320 | 100,657 | 48,957 | 93,410 | 7,039 |
| $R^2$ | 0.188 | 0.141 | 0.341 | 0.300 | 0.264 | 0.268 | x | 0.291 | 0.278 | 0.273 |
| Adj. $R^2$ | 0.184 | 0.101 | 0.245 | 0.213 | 0.258 | 0.227 | x | 0.289 | 0.236 | 0.268 |

The table presents robustness checks of our baseline specification (last column of Table 3). First column reports OLS estimates on a propensity score matched sample. The second and third column exclude the Lombardy region and winter-touristic areas, respectively. In column (4), the sample period is shortened to end on April 4th 2020. In column (5) the definition of "anomaly" in the virus arrival time definition is set to 2 (instead of 1) standard deviations of cumulative excess mortality rate. In column (6) the lagged dependent variable is excluded, while column (7) shows the results of the system GMM estimator, collapsing the instruments matrix (lag2–lag4) and using the two-step technique. Lastly, column (8) shifts the first policy treatment date backwards by ten days, column (9) swaps policy time dummies, and column (10) considers only municipalities in which the virus never circulated during our sample period. Interactions of $d11_t$ with $PopDens_m$, $IncIneq_m$, and $IntMob_m$ are included but suppressed for brevity. $t$ statistics in parentheses. Standard Errors clustered at municipality- and day-level.

*, ** and *** represent statistical significance at the 10%, 5% and, 1% level, respectively. Variable definitions are in S1 Table.

sample (results available on request). Fig 3 presents the propensity score-matched equivalent of Fig 1, and show a graph very similar to the one in Fig 1.

The first column of Table 4 shows the baseline results (equivalent to column (5) of Table 3) for the matched sample. The coefficient of interest is smaller (in absolute terms) but of similar magnitude as in the baseline model (−0.03 versus −0.0358).

Another concern with our analysis is that our results might be driven by specific municipalities, such as the ones in the epicentre of the outbreak in Lombardy or winter tourism hotspots. To investigate this, we focus on subsamples from which we exclude municipalities in Lombardy (column (2)) and winter tourist regions (column (3)). Results are similar to our baseline specification in column (5) of Table 3.

In column (4), we shorten the post-treatment period to April $4^{th}$, that is, before the treatment date for the second policy. This avoids any confounding effect stemming from the second shutdown. The coefficient of interest remains at a similar level.

We also assess the robustness of our baseline result to a more restrictive classification of the virus arrival time. In particular, we identify the onset of the pandemic in a municipality $m$ when the cumulative mortality rate surpasses two (rather than one) standard deviations of its distribution. Column (5) shows that the coefficients on the interaction terms of both policies are very similar compared to the baseline estimates. The results are also robust to changing the "anomaly" threshold to the first decile of each municipality cumulative deaths distribution (results available upon request).

Our main model specification in Eq (1) includes a lagged dependent variable, which is consistent with the idea that new infections (and resulting new deaths) condition highly on

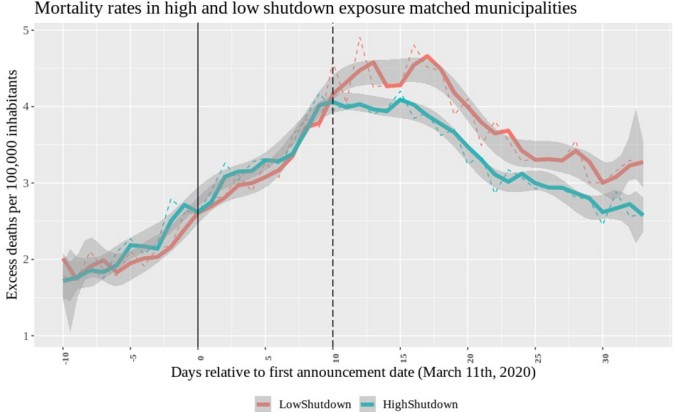

**Fig 3. Mortality rates in high and low shutdown matched municipalities.** 2-step propensity score matched (first on virus arrival week and then on PopDensm, IntMobm and IncIneqm) municipalities' average within-group excess mortality rates over time. Vertical lines identify the first announcement ($\tau = 0$ corresponds to 03/11) and treatment date ($\tau = 10$ to 03/21). A group excess mortality rate is calculated by averaging values across treated municipalities (above the median $Shutdown11m$, N = 873) and across control municipalities (those below the median, N = 444). Blue line: HighShutdown. Red line: LowShutdown.

prevailing infections (and hence recent excess mortality). However, the inclusion of a lagged-dependent variable in OLS regressions creates an econometric problem in small samples, by causing a downward bias in the estimation of the coefficient on the lagged-dependent variable [29]. To rule out any issues resulting from such a bias, we first exclude the lagged dependent variable in column (6). Moreover, in column (7), we estimate a system GMM as in [15], instrumenting the variables in levels with their first differences. We optimally collapse the number of instruments and, to correct for the finite sample standard errors, use a two-step procedure [30]. Both specifications give similar or even stronger results than our baseline specification. Fig 2 confirms this, showing an average point estimate of lives saved exceeding 12,500 over a period of 24 days using the system GMM.

Finally, we consider three falsification tests. First, we move the first policy treatment date to 10 days earlier (column (8)), terminating the sample period at March $20^{th}$. In column (9) we cross-interact time dummies and sectoral exposures, that is, we run the regression with $d25_t \times Shutd11_m$ and $d11_t \times Shutd25_m$. Third, we run the regression for the 477 municipalities in which the virus never circulated at any point during our sample according to our statistical methodology (column (10)). In all placebo tests, the coefficient on the policy interaction term shrinks substantially in size and becomes statistically insignificant.

## Contagion channels

Table 5 further explores the mechanisms behind our baseline results (column (5) of Table 3).

In column (1) we focus on mortality rates of people older than 65 years, that is, among a group that is unlikely to work. Thereby, we effectively exclude employees and business owners from the sample. Column (1) shows that the results – although a bit weaker—are very similar to our baseline specification. We obtain comparable results (unreported) when we focus on people with an age above 80, in which case direct involvement in business activities becomes very unlikely. These results point to a contagion externality from business activities [2], which is an important result from a policy perspective. If people within a firm were predominantly infected, standard economic theory would suggest less of a need for policy interventions as any utility loss due to contagion is then more likely to be internalized (in

**Table 5. Contagion channels.**

| | Elderly | Hospital | Geographic Spillovers | | | Decreasing | | | Sectoral |
| | | | Relative | Absolute | Residents | Effectiveness | | | Decompos. |
| | (1) | (2) | (3) | (4) | (5) | (6) | (7) | (8) | (9) |
|---|---|---|---|---|---|---|---|---|---|
| $d11_t \times Shutd11_m$ | -0.0342*** | -0.0826*** | -0.0284*** | -0.020** | -0.019** | -0.127*** | | -0.0673*** | |
| | (-4.52) | (-4.43) | (-3.47) | (-2.46) | (-2.33) | (-3.84) | | (-4.93) | |
| $d11_t \times Shutd11_m \times Hospit_p$ | | 0.261*** | | | | | | | |
| | | (2.60) | | | | | | | |
| $d11_t \times Shutd11_n$ | | | -0.0276** | -0.109*** | -0.122*** | | | | |
| | | | (-2.31) | (-5.74) | (-6.08) | | | | |
| $d11_t \times Shutd11_m^2$ | | | | | | 0.0023*** | | | |
| | | | | | | (2.97) | | | |
| $d11_t \times Shutd11_m \times Q1_m$ | | | | | | | -0.085*** | | |
| | | | | | | | (-4.31) | | |
| $d11_t \times Shutd11_m \times Q2_m$ | | | | | | | -0.071*** | | |
| | | | | | | | (-3.34) | | |
| $d11_t \times Shutd11_m \times Q3_m$ | | | | | | | -0.007 | | |
| | | | | | | | (-0.41) | | |
| $d11_t \times Food11_m$ | | | | | | | | | -0.0049 |
| | | | | | | | | | (-0.46) |
| $d11_t \times Retail11_m$ | | | | | | | | | -0.0681*** |
| | | | | | | | | | (-3.97) |
| $d11_t \times Personal11_m$ | | | | | | | | | -0.0682** |
| | | | | | | | | | (-2.02) |
| $d25_t \times Shutd25_m$ | -0.00495 | -0.00534 | -0.00397 | 0.00081 | 0.00011 | -0.00652 | -0.0069 | -0.0095** | -0.00613 |
| | (-0.95) | (-1.08) | (-0.79) | (0.19) | (0.04) | (-1.31) | (-1.41) | (-1.99) | (-1.24) |
| $y_{m,t-1}$ | 0.0383*** | 0.0366*** | 0.0366*** | 0.0329*** | 0.0330*** | 0.0355*** | 0.367*** | 0.0306*** | 0.0366*** |
| | (4.16) | (4.07) | (3.91) | (3.68) | (3.71) | (4.06) | (4.05) | (2.83) | (4.06) |
| Interaction Controls | ✓ | ✓ | ✓ | ✓ | ✓ | ✓ | ✓ | ✓ | ✓ |
| Municipality FE | ✓ | ✓ | ✓ | ✓ | ✓ | ✓ | ✓ | ✓ | ✓ |
| ArrDay × Time FE | ✓ | ✓ | ✓ | ✓ | ✓ | ✓ | ✓ | ✓ | ✓ |
| Obs. | 99,466 | 100,563 | 96,853 | 98,729 | 98,713 | 100,563 | 100,563 | 67,363 | 100,563 |
| $R^2$ | 0.267 | 0.270 | 0.270 | 0.272 | 0.231 | 0.270 | 0.269 | 0.270 | 0.270 |
| Adj. $R^2$ | 0.226 | 0.229 | 0.229 | 0.272 | 0.231 | 0.229 | 0.223 | 0.217 | 0.229 |

This table presents results on contagion channels, returns to scale, and other extensions. $Hospit_p$ measures the degree of hospitalization per capita in province $p$. In columns 3, 4 and 5, $Shutd11_n$ are the $1^{st}$ policy exposures of the largest hit municipality within $p$ in relative, absolute and population terms. $Q1 - Q3$ are $Shutd11_m$ tercile dummy variables. The sample in column (8) includes the first two terciles of $Shutd11_m$. $Food11_m$, $Retail11_m$ and $Personal11_m$ are the exposures of $m$ to the food, retail, and personal services, respectively. Interactions of $d11_t$ with $PopDens_m$, $IncIneq_m$, and $IntMob_m$ are included throughout but suppressed for brevity; likewise for pairwise interaction terms in column (2) and (7). $t$-stats in parentheses. S.E. clustered at municipality- and day-level.

*, ** and *** are significance at the 10%, 5% and, 1% level. Variable definitions are in S1 Table.

particular, workers might require higher wages to keep working during the pandemic, or simply stop turning up at work). However, if production also considerably affects mortality rates outside the firm, decentralized production decisions are likely to be sub-optimal, necessitating policy interventions.

Next, column (2) explores the role of hospital capacity. We would expect policies to be more effective in reducing mortality rates in areas with congested hospitals (as virus contagion

is then more likely to be fatal). Consistent with this conjecture, we find the triple interaction coefficient of hospital capacity, the first policy exposure, and the post-treatment dummy to be positive and significant.

So far we have examined the impact of the policy exposure on mortality rates in the municipality itself. However, larger and more developed cities also attract workers from other municipalities. We might therefore expect business shutdowns in large and/or inter-connected municipalities to have positive spillovers on other municipalities. To investigate this hypothesis, we also include in our baseline model an interaction effect of the largest exposed municipality in a province (column (3)). In line with the idea that cities with a high sector concentration attract people from neighbouring municipalities, the coefficient on the interaction between $Shutdown11_n$ and the first shutdown date is negative and significant. Interestingly, it is similar in magnitude as the main within-municipality effect. Similarly, we find statistically significant negative coefficients under two alternative measures: the municipality with the largest absolute number of employees in the shut down sectors (column 4) and with the largest number of inhabitants (column 5). All in all, this points to important spillovers from shutdowns across jurisdictions. [5] show that such cross-jurisdictional spillovers make uncoordinated lockdown decisions inefficient, and derive implications for international cooperation of lockdown policies. Our evidence suggests that coordination among units or centralization of containment policies might be required to achieve optimal containment policies. Fig 2 (last bar) shows that, taking (relative) spillovers into account, the economic effect increases to over 16,000 lives saved.

We also examine whether policy effectiveness exhibits decreasing or increasing returns to scale. First, we include an interaction term with squared shutdown exposure (column (6)). This squared term shows up significantly positive, indicating decreasing returns to scale for business shutdowns. Second, we sort municipalities within each province into terciles according to their exposure to the first policy, and then run a regression interacting our variable of interest with each $Shutdown11_m$ tercile dummy (column (7)). Comparing the coefficients on the shutdown exposure across the different terciles we see that they are consistently declining (in absolute terms) as we move from low to high shutdown exposures. Once again, a marginal unit of shutdown matters less in municipalities with higher shutdown exposure. These results suggest there are declining returns to shutting down businesses. This evidence is consistent with the non-linear nature of epidemiological dynamics. In particular, once the virus is sufficiently contained, the marginal benefit of reducing the reproduction rate further declines. An alternative explanation for the declining coefficients is that a high exposure municipality might also be the largest exposed municipality of the province, in which case the coefficient estimate underestimates the total effect due to the spillovers (as shown previously). However, we still obtain declining coefficients once the largest exposed municipalities are dropped from the sample (results available upon request). In column (8), we formally check whether the declining marginal returns result partially explains the low effectiveness of the second shutdown policy. If shutting down a large part of the economy has already contained the virus, additional shutdowns (on the same day, or two weeks later) matter less. Noticeably, including only the first two $Shutdown11_m$ terciles in the sample, the $d25_t \times Shutdown25_m$ coefficient becomes significant (at 95% level). Therefore, the second shutdown is effective in the low first shutdown exposure sample and hence, our result of a lower effectiveness of the second shutdown is driven by decreasing returns to scale.

The last column of the table decomposes the first shutdown exposure into different sectors. We create exposure variables for all three sectors (food, retail, and personal services) following the same approach as for the total exposure ($Shutdown11_m$) variable. The results show that all individual exposures obtain a negative coefficient (the coefficient on food is insignificant,

though). Retail and personal services obtain similar coefficients, which are negative and strongly significant (both statistically and economically). These results inform policy makers about the potential benefits of shutting down specific sectors of the economy. In particular, it points to relatively high benefits of shutting down retail activities. In fact, on top of shutdowns being fairly effective there, brick-and-mortar retail has a close substitute (online shopping) and hence, its shutdown might cause a lower loss of consumer welfare.

## Conclusion

This paper has examined the impact of national business shutdowns in Italy during the Covid-19 crisis. Employing a difference-in-differences approach we have found that municipalities more exposed to shutdowns experience subsequently lower mortality rates. This suggests that business closures are effective in containing the spread of a virus and, ultimately, save lives. The effects are economically large and point to significant benefits from business closure orders during a pandemic. Our analysis suggests rapidly declining (marginal) benefits from shutdowns and points to business decisions having important (contagion) spillovers, on individuals outside the business as well as individuals in other localities. Our findings provide valuable information to policy-makers involved in navigating the Covid-19 crisis, as well as for managing future pandemics.

## Supporting information

**S1 Table. Variable definition.** This table shows the definition of each variable used in the empirical analysis.
(PDF)

## Author Contributions

**Conceptualization:** Dion Bongaerts, Francesco Mazzola, Wolf Wagner.

**Data curation:** Francesco Mazzola.

**Formal analysis:** Dion Bongaerts, Wolf Wagner.

**Methodology:** Dion Bongaerts, Francesco Mazzola.

**Project administration:** Wolf Wagner.

**Software:** Francesco Mazzola.

**Supervision:** Dion Bongaerts, Wolf Wagner.

**Validation:** Francesco Mazzola.

**Visualization:** Francesco Mazzola.

**Writing – original draft:** Dion Bongaerts, Wolf Wagner.

**Writing – review & editing:** Dion Bongaerts, Wolf Wagner.

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
