## [Decision Letter · Decision Letter 0]

12 Feb 2021

PONE-D-20-36331

Closed For Business

PLOS ONE

Dear Dr. Mazzola,

Thank you for submitting your manuscript to PLOS ONE. After careful consideration, we feel that it has merit but does not fully meet PLOS ONE’s publication criteria as it currently stands. Therefore, we invite you to submit a revised version of the manuscript that addresses the points raised during the review process.

The manuscript approaches a very topical subject, but further robustness checks are required.

We look forward to receiving your revised manuscript.

Kind regards,

Stefan Cristian Gherghina, PhD. Habil.

Academic Editor

PLOS ONE

Journal Requirements:

2. Please modify the title to ensure that it is meeting PLOS’ guidelines (https://journals.plos.org/plosone/s/submission-guidelines#loc-title). In particular, the title should be "specific, descriptive, concise, and comprehensible to readers outside the field" and in this case it is not informative and specific about your study's scope and methodology.

Reviewers' comments:

Reviewer's Responses to Questions

**Comments to the Author**

1. Is the manuscript technically sound, and do the data support the conclusions?

Reviewer #1: Yes

Reviewer #2: Yes

2. Has the statistical analysis been performed appropriately and rigorously? 

Reviewer #1: Yes

Reviewer #2: Yes

3. Have the authors made all data underlying the findings in their manuscript fully available?

Reviewer #1: Yes

Reviewer #2: Yes

4. Is the manuscript presented in an intelligible fashion and written in standard English?

Reviewer #1: Yes

Reviewer #2: Yes

5. Review Comments to the Author

Reviewer #1: I have attached my review as an attachment along with this review submission, where I have mentioned my comments.

It is a very well written article. It needs minor revisions for some grammatical errors.

Reviewer #2: Review of “Closed for Business”

Recommendation:

Summary:

“Closed for Business” uses a differences-in-differences technique to analyze data from Italy’s lockdowns amidst COVID-19 to evaluate the effectiveness of the lockdowns in saving lives. The paper takes advantage of differences in the sectoral distribution of employment between municipalities to evaluate whether those areas that were most exposed to sector-specific lockdowns saw a decline in their mortality rates. In brief, the answer is “yes”. In addition, the paper shows that there are diminishing marginal returns to lockdowns (that is, that locking down twice as much of the economy is less than twice as effective at slowing the spread), and that significant spillovers between business centers and nearby municipalities, suggesting there are benefits to coordinating lockdown policies.

General Evaluation:

I found this to be an interesting, timely paper with interesting policy implications. The methods are generally well-defended. However, there are some points where I believe additional explanation or slight changes in method would be warranted, discussed in the section below.

Recommendations:

I have divided by recommendations into “major” recommendations and “minor” recommendations, based on my view of their importance.

Major Recommendations

1. Page 7 – The paper says that shutdown exposure is constructed by “dividing the number of employees and employers in sectors shut down… by the total number of employees and employers in the municipality.” It’s unclear why employees are being added to employers, as this seems to be mixing together two very different units (people and firms). If there are differences in typical firm sizes between municipalities and if shutdowns have a disparate impact on firms of a specific size, this could lead to some strange behavior in this variable, as the number of firms would be heavily weighted in importance in municipalities with very small firms and receive less weight in municipalities with fewer, but larger firms. I recommend constructing the Shutdown variables on the basis of employees alone (or at least using this alternative construction as a robustness check).

2. Page 10 – The paper defines the business centre as “the municipality with the highest share of closed sectors among the province’s larger municipalities (with at least 16,500 inhabitants).” It’s unclear why a high percentage of closed sectors would imply that a municipality is a business centre. Why not the municipality with the greatest number of inhabitants? Or the greatest number of employees (rather than share of employees) in the shutdown sectors? Explaining the reasoning behind this construction would be helpful.

Minor Recommendations

1. Page 2 – first paragraph - Was Sweden’s approach really “laissez-faire”? My understanding is that they did make some attempt to isolate the most at risk populations and to restrict the sizes of gatherings.

2. Page 6 – the phrase “Due to dispersion” is unclear to me. I’m guessing it just means that there will be variation around the median, but, in the context of an epidemic, I wonder if “dispersion” has an epidemiological meaning that I’m not familiar with.

3. Page 6 – Equation 1 – I’m not sure why there aren’t parameters put on the dummy variables for d11t and d25t. It makes no difference to the substance.

4. On page 11-12 – The paper describes the construction of Figure 1 by discussing dividing municipalities into cohorts based on their own stage in the pandemic. Then, “Within each of these cohorts, we calculate the average excess mortality rates for municipalities above and below the median shutdown exposure, and then average across cohorts.” This process seems like the only effect it would have is overweighting the small cohorts, it’s not obvious how this corrects for the stage of the pandemic. More explanation would be helpful.

5. On page 20, near the end of the first paragraph – the “Shutdown25” has inconsistent subscript use between the 2 and 5. I think it should be “Shutdown25” with just an “m” in the subscript.

---

## [Author Response · Author response to Decision Letter 0]

22 Apr 2021

Dear Editor, Dear Referees,

Thank you so much for taking the effort to carefully read our paper and provide us with thoughtful and constructive comments and suggestions. We are excited to have had the opportunity to revise our paper for potential publication in PLOS ONE. Over the last weeks, we have incorporated all your comments into an improved version of the paper. In the attached files "Response to reviewers", we provide a detailed response to your comments separately. For our responses, we repeated your comments (in italics) and carefully indicated how we addressed each of your comments and where in the paper these changes can be found (in regular font). All references to pages and sections are based on the new, revised version of the paper, unless indicated otherwise.

Best regards,

The authors

---

## [Decision Letter · Decision Letter 1]

26 Apr 2021

Closed For Business: the mortality impact of business closures during the Covid-19 pandemic

PONE-D-20-36331R1

Dear Dr. Mazzola,

We’re pleased to inform you that your manuscript has been judged scientifically suitable for publication and will be formally accepted for publication once it meets all outstanding technical requirements.

Kind regards,

Stefan Cristian Gherghina, PhD. Habil.

Academic Editor

PLOS ONE

Additional Editor Comments (optional):

Reviewers' comments:

Reviewer's Responses to Questions

**Comments to the Author**

1. If the authors have adequately addressed your comments raised in a previous round of review and you feel that this manuscript is now acceptable for publication, you may indicate that here to bypass the “Comments to the Author” section, enter your conflict of interest statement in the “Confidential to Editor” section, and submit your "Accept" recommendation.

Reviewer #1: All comments have been addressed

Reviewer #2: All comments have been addressed

2. Is the manuscript technically sound, and do the data support the conclusions?

Reviewer #1: Yes

Reviewer #2: Yes

3. Has the statistical analysis been performed appropriately and rigorously? 

Reviewer #1: Yes

Reviewer #2: Yes

4. Have the authors made all data underlying the findings in their manuscript fully available?

Reviewer #1: Yes

Reviewer #2: Yes

5. Is the manuscript presented in an intelligible fashion and written in standard English?

Reviewer #1: Yes

Reviewer #2: Yes

6. Review Comments to the Author

Reviewer #1: No comments. authors seem to have incorporated all the needed changes. I think they have done a great job !

Reviewer #2: Than you very much to the authors for addressing the comments on the original version so thoroughly. I am satisfied with the current version of the paper.

7. PLOS authors have the option to publish the peer review history of their article (what does this mean?). If published, this will include your full peer review and any attached files.

Reviewer #1: No

Reviewer #2: No

---

## [Editor Report · Acceptance letter]

4 May 2021

PONE-D-20-36331R1 

Closed for Business: the mortality impact of business closures during the Covid-19 pandemic 

Dear Dr. Mazzola:

I'm pleased to inform you that your manuscript has been deemed suitable for publication in PLOS ONE. Congratulations! Your manuscript is now with our production department. 

Kind regards, 

on behalf of

Dr. Stefan Cristian Gherghina 

Academic Editor

PLOS ONE